# The Process of Soil Nutrient Stabilization in Micro-Patches in Alpine *Kobresia* Meadows

Li Lin [1,2], Guangmin Cao [1,*], Xiaowei Guo [1,*], Qian Li [1], Dawen Qian [1], Yangong Du [1], Junjie Huang [3], Bo Fan [1], Bencuo Li [1], Yuting Lan [1] and Mengke Si [1]

[1] Northwest Institute of Plateau Biology and Chinese Academy of Sciences, Xining 810008, China
[2] University of Chinese Academy of Sciences, Beijing 100049, China
[3] School of Mechanical and Power Engineering, Henan Polytechnic University, Jiaozuo 454000, China
[*] Correspondence: caogm@nwipb.cas.cn (G.C.); guoxw@nwipb.cas.cn (X.G.)

**Abstract:** Micro-patches are the basic unit of grazing ecosystems; the characteristics of these micro-patches are relatively stable in species under different grazing intensities in the same vegetation, but obviously different in terms of the distribution pattern. This leads to differentiation of plant community numerical characteristics under different grazing intensities. Understanding the driving force of soil nutrient variation in micro-patches under grazing disturbances will help us comprehend the regulation strategy and adaptation mechanisms of the ecosystem against over-disturbance. We designed four scales: spatial (three typical micro-patches), temporal serial (6 years), a degradation succession process (four key degradation stages), and recovery treatment (three treatments: the original grazing intensity based on herder preferences, half of the original grazing intensity, and zero grazing). The soil nutrient characteristics used to estimate stabilization were the typical soil total nutrient content (soil organic matter [SOM], total nitrogen [TN], total carbon [TC], inorganic carbon [IC], total phosphorus [STP], total potassium [TK], and pH), and available soil nutrients ($NH_4^+$, $NO_3^-$, phosphorous [avP], and potassium [avK]). Variations in the SOM, TC, IC, TN, STP, avK, and $NO_3^-$ levels in the main root distribution layers (0–20 cm) on the spatial scale were 69.8–79.7%, 61.4–80.35%, 49.8–79.58%, 60.52–76.34%, 46.44–89.89%, 45.5–71.36%, and 59.21–65.38%, respectively, which accounted for the largest variation in the four scales, based on multivariable analysis. The variations in the avP and $NH_4^+$ content of the main root distribution layers (0–20 cm) at the temporal scale were 46.42–67.93% and 48.11–64.55%, respectively, which accounted for the greatest variation in the four scales, based on a multivariable analysis. Upon comparing the degradation succession stages and recovery treatment in each stage, we found that the variation in avP, avK, STP, TN, TC, SOM, TC, and TN content was greater at the degradation succession scale than at the recovery treatment scale. The soil nutrient content of the micro-patches exhibited the smallest decrease in the Gramineae-*Kobresia* transformation (G-KP) micro-patch, followed by the Gramineae micro-patches (G) and *Kobresia* micro-patches (KP). The number of G micro-patches decreased with increasing grazing intensity whereas the number of KP micro-patches increased. When the number of KP micro-patches increased to a certain degree, the number of G-KP micro-patches then increased as well. G-KP micro-patches, characterized by cracking in the mattic epipedon in alpine meadows, increased with the grazing intensity increasing in a certain degree in *K. pygmaea* meadows with mattic epipedon cracking (CP); the latter buffered the nutrient variation and maintained the soil nutrients' relative stability in the ecosystem. Thus, CP formed the buffer stage for maintaining self-stabilization during a regime shift and was considered the withstanding stage during the alpine *Kobresia* meadow degradation process.

**Keywords:** ecosystem stabilization; soil nutrient variation; alpine meadow management; warming stage; degradation process

## 1. Introduction

The alpine *Kobresia* meadow is the main body of the grazing ecosystem on the Qinghai-Tibet Plateau (QTP), which is of great importance for maintaining the ecological security, the Tibetan culture, etc. In the past several decades, the characteristics of composition have obviously varied even within topographic and climatic areas with the same grazing intensity. Previous studies showed that grazing intensity is the main factor that controls the differences in characteristics of the plant community and soil nutrient content in a given geographical and climatic area [1]. The difference in grazing intensity is also a key factor responsible for the coexistence of multiple steady states within the same geographical and climatic area under different disturbance intensities [1,2]. The typical succession stages of alpine meadows along with increasing grazing intensities are Gramineae-*Kobresia* meadows, *Kobresia* meadows, and "black-beach soil" meadows, which are full of annual and biennial forb in summer, uncovered vegetation in winter, and look like a black beach in winter. The latter two stages are considered dis-climax zonal vegetation communities under overgrazing pressure, based on the internal and external coupling hypotheses of the alpine *Kobresia* meadow degradation succession process [3,4]. The degradation of Alpine *Kobresia* meadows caused by overgrazing is often accompanied by reduced plant community production and ecological service. Therefore, clarifying the succession law of degraded grassland is of great importance in formulating management strategies and carrying out the restoration of degraded meadows.

Gramineae (G) micro-patches, *Kobresia* (KP) micro-patches, and Gramineae-*Kobresia* transformation (G-KP) micro-patches are the typical micro-patches observed during *Kobresia* meadow succession [3] and may coexist in several succession stages [1]. In general, a micro-patch is the basic constituent of plant communities and soil nutrients, originating from habitat fragmentation at the spatial scale [5]. Combinations of different micro-patches can generate micro-habitats for many species [6–9], whereas changing the number and type of micro-patches can change the patterns and structure of plant communities in the ecosystem [10,11]. This would in turn adjust the nutrient utilization model at different temporal and spatial scales, increasing the stability of the entire ecosystem [12] and even pushing it toward regime shifts [13]. The characteristics of ecosystem components at the micro-patch scale vary frequently over a short time span, but remain stable over a relatively long time span at the large spatial scale via positive and negative feedback of different micro-patches to disturbance [14]. This causes the characteristics of the whole ecosystem to behave in oscillating, volatile, nonlinear, or stable manners, depending on the intensity of the disturbance [13,15,16]. Because the characteristics of micro-patches can reflect and explain the survival strategy model and construction patterns, they have been used to estimate the self-stability and regime shift of ecosystems [17–19].

Soil nutrients and plants form the basic components of micro-patches. The characteristics of soil nutrients in micro-patches, especially the total soil nutrient content, have a strong relationship with the plant community. Differences in the micro-climate and micro-topography and heterogeneity of disturbance intensity determine the heterogeneity of the habitat, provide basic stability to biotic components of the ecosystem during disturbances, and maintain components of the entire ecosystem in a relative steady state during disturbances. Thus, the soil nutrients in a micro-patch can reflect its suitability for typical local plants and vice versa [20–22]. In comparison, plants can find and colonize suitable micro-patches, possibly changing the composition and pattern of the plant community in the ecosystem [23] and leading to regime shifts during the plant community succession process [24]. From this perspective, soil nutrients play an important role in maintaining ecosystem components in a steady state [25–28]. Variations in the amount and stability of soil nutrients can reflect the history and characteristics of micro-patches and plant community succession [3,14].

In general, it easier to recognize micro-patches by plant community than by soil nutrient composition, so less attention has been paid to soil nutrients. However, in practice, we do not know how soil nutrients differ within a given micro-patch type across different

stages of the overgrazing-induced degradation succession process. The grazing pasture in the QTP has complex topography, climatic conditions, and grazing intensities, with the characteristics of the micro-patch simultaneously controlled temporally and spatially by herd management. We also do not know the individual contributions of these factors (temporal, spatial, grazing intensity) to the variation in soil nutrients. Moreover, the threshold stage, based on the soil nutrient content, in a multiple-steady-state regime shift process is unknown. Therefore, we carried out a series of studies to answer these questions. We used 11 soil nutrient indices ($NH_4^+$, $NO_3^-$, available phosphorus content [avP], available potassium [avK], soil organic matter [SOM] content, total soil nitrogen [TN] content, soil inorganic carbon [IC] content, total soil carbon [TC] content, total soil phosphorus [STP] content, total soil potassium [TK] content, and soil pH) and estimated their stabilization across four scales. The four scales, in the four key stages of the alpine *Kobresia* meadow degradation succession process, are the degradation process scale, three typical micro-patches (G, KP, and G-KP micro-patches) as the spatial scale, three grazing intensities (raw grazing intensity based on one herd previously used in the pastures, half of the raw grazing intensity, and zero grazing intensity) as the recovery scale, and 6 years (from 2013 to 2018) of continuous monitoring data as the temporal scale. We hypothesized that the soil nutrient contents and their stabilization in typical micro-patches vary both in spatial and temporal aspects based on the degradation process and recovery process scales and that the factor with greatest variation with respect to the total variation in soil nutrients is a driver of such variation. The aims of this study were to describe the process of variation in soil nutrient content in typical micro-patches throughout the degradation and recovery processes and to determine the withstanding stage based on the amount of soil nutrient variation need to maintain self-stability. The findings of this study will provide theoretical insights into vegetation succession and the selection of appropriate adaptive management strategies for the recovery of degradation in alpine meadows.

## 2. Materials and Methods

### 2.1. Study Area and Sampling Setting

We selected four key stages in the degradation succession process of the alpine *Kobresia* meadows in the Huangcheng Village of Menyuan County, Haibei Prefecture, Qinghai Province, China (Figure 1). Our experiment was divided into four statistical scales. The first scale was temporal, for which we used 6 years of physical data for pH and SOM, TC, IC, TN, STP, TK, avK, and $NH_4^+$ and $NO_3^-$ content. The second scale was a degradation succession process scale, which comprised Gramineae-*K.humilis* (GK) meadows, *K. humilis* meadows (KH), *Kobresia pygmaea* meadow with mattic epipedon thickening (TP), and *K. pygmaea* meadow with mattic epipedon cracking (CP) based on the Qinghai Province Local Standard (DB/63T 1413-2015 and DB/63T 1414-2015) (Table 1). The third scale was the recovery treatment scale, which comprised the original grazing intensity previously used by pasture managers (referred to as free), half of the original grazing intensity (referred to as reduced), and zero grazing intensity (referred to as prohibited). The final scale was the spatial scale, which comprised KP micro-patches, G-KP transformational micro-patches, and G micro-patches (G) (Table 2 and Figure 2).

**Table 1.** General information about the sampling area.

|  | Gramineae-*Kobresia humilis* Meadow | *Kobresia humilis* Meadow | *Kobresia pygmaea* Meadow with Mattic Epipedon Thickening | *Kobresia pygmaea* Meadow with Mattic Epipedon Cracking |
|---|---|---|---|---|
| Location | 37°39.023′ N, 101°10.638′ E; 3230 m | 37°40.155′ N, 101°10.021′ E; 3241 m | 37°40.054′ N, 101°10.620′ E; 3239 m | 37°42.089′ N, 101°15.928′ E; 3278 m |
| Grazing intensity (number of sheep/ha) | 3.75 | 7.25 | 8.50 | 11.75 |
| Soil surface features | No obvious cracking | No obvious cracking | Cracking area less than 8% | Cracking areas 10–15% |

**Table 1.** *Cont.*

|  | Gramineae-*Kobresia humilis* Meadow | *Kobresia humilis* Meadow | *Kobresia pygmaea* Meadow with Mattic Epipedon Thickening | *Kobresia pygmaea* Meadow with Mattic Epipedon Cracking |
|---|---|---|---|---|
| **Thickness of the mattic epipedon** | <5 cm | 5–6 cm | 7–8 cm | 8–9 cm |

Adapted with permission from Lin et al. [2].

**Table 2.** General information about the micro-patches.

|  | **Gramineae Micro-Patch** | **Gramineae-*Kobresia* Transformation Micro-Patch** | ***Kobresia* Micro-Patch** |
|---|---|---|---|
| **Characteristics of the plant community** | *Stipa* spp., *Festuca* spp., *Ptilagrostis* spp. and *Helictotrichon tibeticum* individually or in combination as the dominant plants, with *Kobresia humilis* and/or *K. pgymaea* as the companion species | *K. humilis*, *K. pgymaea*, *Stipa* spp., *Festuca* spp., *Ptilagrostis* spp. and *Helictotrichon tibeticum* individually or in combination as the dominant plants | *K. humilis* or *K. pgymaea* as the dominant plants, with *Stipa* spp., *Festuca* spp., *Ptilagrostis* spp. and *Helictotrichon tibeticum* as the companion species |
| **Characteristics of the soil surface** | No obvious cracking in the mattic epipedon | Obvious cracking in the mattic epipedon across the micro-patch, with | No obvious cracking in the mattic epipedon |
| **Characteristics of the soil biological crust** | Moss as the dominant soil biological crust, with algae and lichen as the companion biological crust | moss, algae, and lichen as the dominant biological soil components | Algae and lichen as the dominant soil biological crust, with moss as the companion biological crust |

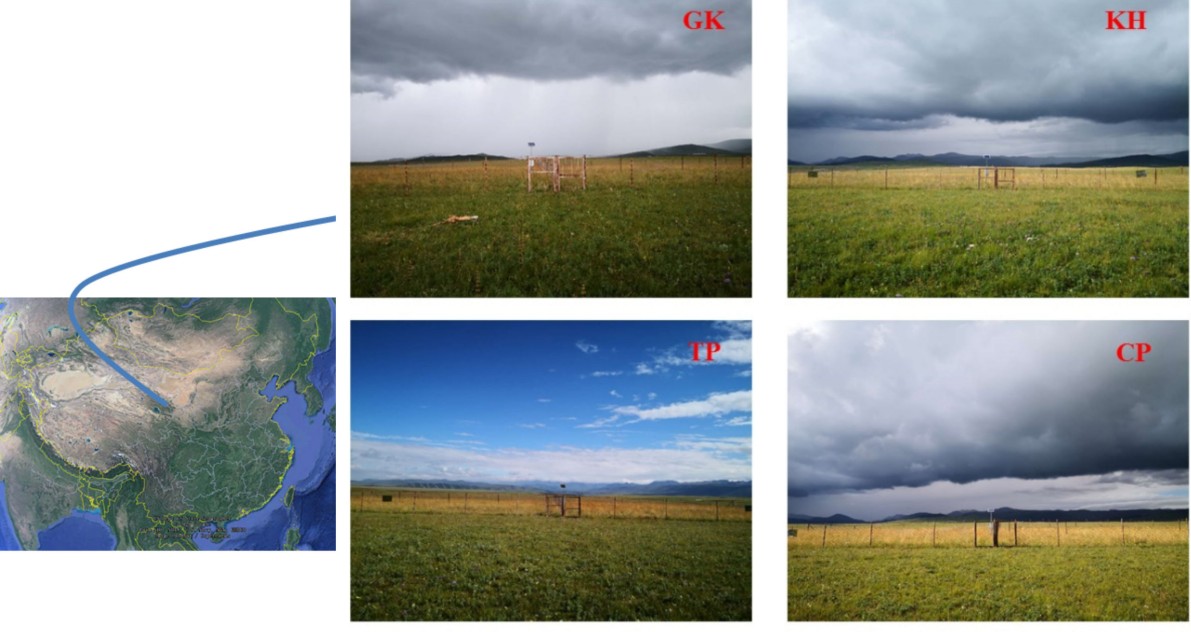

**Figure 1.** The sampling plots. Note: (**GK**), Gramineae-*K. humilis* meadow; (**KH**), *K. humilis* meadow; (**TP**), *K. pygmaea* meadow with mattic epipedon thickening; (**CP**), *K. pygmaea* meadow with mattic epipedon cracking. Adapted with permission from Lin et al. [2].

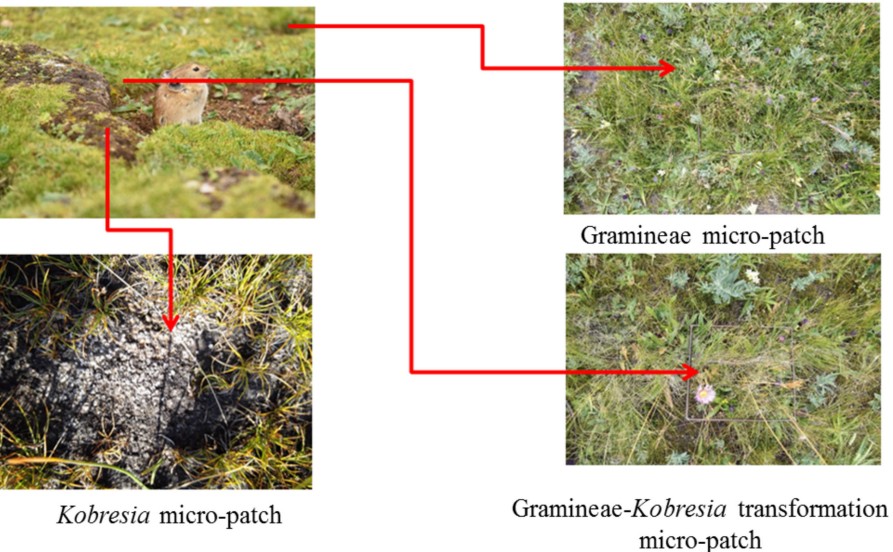

Figure 2. Typical micro-patches seen in the research plots.

*2.2. Sample Collection and Nutrient Analysis*

Soil samples were collected with a soil auger at depths of 0–5, 5–10, 10–20, 20–30, and 30–40 cm. Six replicate samples were collected from each soil layer. The sampling times were at the end of August or beginning of September, from 2013 to 2018. All of the soil samples were air-dried and filtered through 2 mm and 0.25 mm sieves after removing the stones and roots. The 2-mm-filtered soil samples were used to analyze the available soil nutrients and pH, whereas the 0.25-mm-filtered soil samples were used to analyze the total soil nutrient content. The available soil nutrients were nitrate nitrogen ($NH_4^+$), ammonium nitrogen ($NO_3^-$), avP, and avK content. The total nutrients were SOM, TN, IC, TC, and STP content and pH.

The concentrations of TC and TN in the soil were determined by dry combustion using an Elemental Analyzer (Elementar Vario EL Cube, Elementar Inc., Hesse, Germany). IC concentration was analyzed via a gasometric method (Eijkelkamp Calcimeter, Eijkelkamp Instruments Inc., EN Giesbeek, The Netherlands). Soil $NH_4^+$ and $NO_3^-$ concentrations were analyzed by colorimetry using a Discrete Chemistry Analyzer (Smartchem 140, WESTCO Scientific Instruments Inc., Brookfield, CT, USA) [2]. AvP concentrations were analyzed by sodium bicarbonate extraction and molybdenum antimony colorimetry (GB 12297-90). AvK concentrations were analyzed by ammonium acetate extraction flame photometry (GB 7856-87). pH concentrations were analyzed using a potentiometric method (GB 7859-87). STP concentrations were analyzed by acid soluble molybdenum antimony colorimetry (GB 7852-87). SOM concentrations were analyzed using the formula of SOM = (TC−IC) × 1.724 (GB 7857-87) [29].

*2.3. Statistical Analysis*

The variability in soil nutrient content across the temporal, spatial, recovery treatment, and plant community succession process scales was analyzed by multivariable analysis [30]. Interactions between soil nutrient content across the four scales were analyzed using SPSS (v19.0; IBM, Armonk, NY, USA) using a general linear model with a significance level of 0.05. Dissimilarity was based on the formula: dissimilarity (coefficient of variation [CV]) = standard deviation/mean × 100% [31,32]. We used SOM as an example to calculate the variation trends in micro-patches. In this part of the study, basic soil nutrient data from six repeated measurements obtained over the sampling season were used to generate one basic datum for each plot and 6 years of data were used to calculate the variation in soil nutrient content across the degradation process stages. The variation in soil nutrient content at the temporal scale was calculated based on repeated measurements obtained

over a period of 1 year and one plot was generated for the basic data (A). The ratio of the micro-patch area to the total plot area was used as the weighted ratio (W). The soil nutrient content in one research plot over 1 year was calculated as follows: $CW = A_1 \times W_1 + A_2 \times W_2 + A_3 \times W_3$ [33].

To calculate the weight ratio of micro-patch area in each year, we used three transects (each with a length of approximately 100 m) in different directions, recorded the distance of each micro-patch on the transecting, added the lengths of the same micro-patch in one transect, divided this by the total length, and used the resulting value as the basic weight ratio. The mean of the three weight ratios was then calculated as the final weight ratio (W).

## 3. Results

### 3.1. Variation in Soil Nutrient Content in an Alpine Meadows at Different Succession Stages

The main factors affecting the soil nutrient content of alpine meadows in the same geographical and climatic conditions were sampling year (temporal scale), succession stage (degradation process scale), grazing intensity (recovery scales), and plant community patches (spatial scale). With the exception of STP content in the grazing × soil layer, interannual × grazing intensity × soil layer, and plot × patchiness × soil layer, which had $\alpha$ values of 0.489, 0.353, and 0.206, respectively, these factors had interaction effects.

The spatial scale (micro-patches) made a more significant contribution than the temporal scale to the variability in total soil nutrient content (such as SOM, TN, TC, and STP). The available nutrients in the soil (such as avP and $NH_4^+$) were more sensitive at the temporal scale than at the spatial scale.

In the plant community succession scale, the pH value and SOM, TN, TC, IC, STP, avK, and avP content were more stable at the recovery process scale than at the degradation process scale, whereas $NH_4^+$ and $NO_3^-$ content were more sensitive at the recovery process scale than at the degradation process scale. (Table 3).

**Table 3.** Multivariable analysis of soil nutrient characteristics.

| | | 0–5 cm | 5–10 cm | 10–20 cm | 20–30 cm | 30–40 cm |
|---|---|---|---|---|---|---|
| **pH** | Spatial | 25.30 | 26.02 | 0.53 | 0.28 | 0.28 |
| | Recovery treatment | 0.00 | 0.00 | 18.75 | 8.53 | 6.95 |
| | Degradation succession | 35.27 | 38.89 | 28.56 | 10.93 | 12.49 |
| | Temporal | 39.43 | 35.09 | 52.16 | 80.26 | 80.29 |
| **SOM** | Spatial | 79.70 | 69.80 | 78.48 | 80.51 | 58.34 |
| | Recovery treatment | 0.25 | 0.00 | 0.00 | 0.00 | 2.07 |
| | Degradation succession | 9.89 | 10.07 | 9.77 | 8.10 | 33.17 |
| | Temporal | 10.16 | 20.13 | 11.75 | 11.39 | 6.42 |
| **TC** | Spatial | 80.35 | 72.37 | 61.44 | 20.23 | 12.45 |
| | Recovery treatment | 0.00 | 0.00 | 0.00 | 0.00 | 0.62 |
| | Degradation succession | 10.49 | 11.26 | 12.50 | 4.62 | 7.22 |
| | Temporal | 9.16 | 16.37 | 26.06 | 75.16 | 79.71 |
| **IC** | Spatial | 49.80 | 66.63 | 79.58 | 96.83 | 53.29 |
| | Recovery treatment | 20.99 | 0.00 | 0.00 | 0.00 | 0.00 |
| | Degradation succession | 5.39 | 4.85 | 17.84 | 3.17 | 4.81 |
| | Temporal | 23.82 | 28.52 | 2.58 | 0.00 | 41.90 |
| **TN** | Spatial | 76.34 | 60.52 | 78.32 | 64.45 | 74.40 |
| | Recovery treatment | 6.38 | 2.27 | 0.00 | 8.93 | 9.19 |
| | Degradation succession | 13.70 | 19.37 | 5.40 | 7.55 | 4.99 |
| | Temporal | 3.58 | 17.84 | 16.29 | 19.07 | 11.43 |
| **STP** | Spatial | 80.15 | 89.89 | 46.44 | 53.82 | 36.46 |
| | Recovery treatment | 0.00 | 0.00 | 44.99 | 3.96 | 17.16 |
| | Degradation succession | 19.85 | 10.11 | 0.93 | 12.48 | 24.23 |
| | Temporal | 0.00 | 0.00 | 7.64 | 29.74 | 22.15 |

**Table 3.** *Cont.*

|  |  | 0–5 cm | 5–10 cm | 10–20 cm | 20–30 cm | 30–40 cm |
|---|---|---|---|---|---|---|
| **TK** | Spatial | 42.28 | 31.44 | 31.77 | 31.17 | 25.37 |
|  | Recovery treatment | 39.57 | 33.62 | 24.61 | 0.00 | 17.10 |
|  | Degradation succession | 3.96 | 24.82 | 15.21 | 26.62 | 10.95 |
|  | Temporal | 14.18 | 10.12 | 28.41 | 42.20 | 46.58 |
| **avK** | Spatial | 45.50 | 64.99 | 71.36 | 65.71 | 64.25 |
|  | Recovery treatment | 1.83 | 0.00 | 0.00 | 0.00 | 0.00 |
|  | Degradation succession | 17.99 | 32.79 | 27.99 | 19.24 | 7.55 |
|  | Temporal | 34.68 | 2.22 | 0.66 | 15.05 | 28.20 |
| **avP** | Spatial | 23.44 | 14.79 | 30.28 | 40.41 | 24.56 |
|  | Recovery treatment | 0.00 | 0.00 | 0.23 | 1.79 | 8.88 |
|  | Degradation succession | 30.14 | 17.28 | 17.74 | 28.05 | 15.68 |
|  | Temporal | 46.42 | 67.93 | 51.75 | 29.75 | 50.88 |
| **$NH_4^+$** | Spatial | 27.00 | 18.37 | 35.45 | 26.64 | 26.29 |
|  | Recovery treatment | 19.80 | 33.52 | 0.00 | 4.53 | 10.78 |
|  | Degradation succession | 4.81 | 0.00 | 0.00 | 0.00 | 0.00 |
|  | Temporal | 48.39 | 48.11 | 64.55 | 68.83 | 62.92 |
| **$NO_3^-$** | Spatial | 59.21 | 60.15 | 65.38 | 48.81 | 44.21 |
|  | Recovery treatment | 26.33 | 16.83 | 3.59 | 21.89 | 38.94 |
|  | Degradation succession | 0.00 | 0.00 | 0.00 | 0.00 | 0.00 |
|  | Temporal | 14.46 | 23.02 | 31.03 | 29.31 | 16.86 |

Note: SOM, soil organic matter; TC, total carbon; IC, inorganic carbon; TN, total nitrogen; STP, total phosphorous; TK, total potassium; avK, available potassium; avP, available phosphorous. The values are given as percentages in the table. The Gramineae-*Kobresia* transformation, Gramineae, and *Kobresia* micro-patches are at the spatial scale.; raw grazing intensity based on one herd previously used in the pastures, half of the raw grazing intensity, and zero grazing intensity values are at the recovery treatment scale; the Gramineae-*K. humilis* meadow, *K. humilis* meadow, *K. pygmaea* meadow with mattic epipedon thickening, and *K. pygmaea* meadow with mattic epipedon cracking degradation succession stage values are at the degradation succession scale; and the 2013–2018 6-year scale values are at the temporal scale. The soil nutrient contents were calculated as the average of the soil nutrient contents in one year in the every soil layers. Units are given as %.

SOM content is closely related to the content of most soil nutrients (such as TC and TN). It can be used as an indicator of the total nutrient content of soil in soil fertility studies. Therefore, we used SOM content as a representative metric to estimate the characteristics of the total soil nutrient content during the plant community degradation succession process. Multivariable analysis showed that the variation in SOM content at the temporal scale explained 6.42–20.13% of the total variation, which was higher than the amount of variation explained by the recovery treatment scale (<2.07%). However, the variation at these two scales was lower than the variation in SOM content at the micro-patch scale (58.34–80.51%). Thus, we inferred that the total nutrient content of the soil was more affected by the degradation stage than by the micro-patch. Short-term recovery treatment could not change the total nutrient content of the soil (Table 3).

This research showed that avP, $NH_4^+$, and $NO_3^-$ are all available soil nutrients in alpine meadows and are more prone to change than the total soil nutrient contents at the four scales studied. The variation in avP content at the temporal scale explained 29.75–67.93% of the total variation. The variation in avP content was higher at the plant community degradation succession scale than at the spatial scale at a depth of 0–10 cm, but higher at the spatial scale at a depth of 10–40 cm. The lowest variation was seen for recovery treatments, which showed < 8.88% variation in all soil layers (Table 3).

$NH_4^+$ is another available soil nutrient in alpine meadows. At the temporal and spatial scales, for example, the variation in $NH_4^+$ content was higher at the temporal scale than at the spatial scale; compared with the recovery and degradation processes, the recovery treatment scale had a higher effect on $NH_4^+$ content variation than the degradation process (Table 3).

$NO_3^-$ is yet another available soil nutrient in alpine meadows. The spatial scale showed the greatest variation in $NO_3^-$ content among all the scales. The variation in $NO_3^-$ content was higher at the recovery treatment scale than at the degradation scale (Table 3).

We concluded that the soil nutrients had different reactions to different scales. The available soil nutrient content was more prone to change at the temporal scale. Reducing the grazing intensity for a short time increased the available content of some of the nutrients in the soil (such as $NO_3^-$), but had no effect on the total soil nutrient content. Therefore, the variation in total nutrient content was primarily affected by the micro-patch, followed by the plant community degradation succession process. Reducing the grazing intensity for a short of time had no significant effect on the total nutrient content. However, the available soil nutrient content, especially the avP and $NO_3^-$ content, was significantly affected by recovery treatment. The available nutrient content in soil was sensitive to a reduction in grazing intensity, regardless of the degenerative succession process stage, whereas the soil total nutrient content was sensitive to the plant community succession stage, regardless of the grazing disturbance intensity (Table 3).

### 3.2. Variation in Total Soil Organic Matter in Micro-Patches

The largest contribution to SOM content variation was observed at the micro-patch scale, with the greatest variation observed in the 0–5 cm and 5–10 cm soil layers. Taking the 0–5 cm layer as an example, the differences (CV) between the KP and G-KP micro-patches were 8.9–16.7%, 5.4–23.6% and 1.7–23.2% for free-grazing, reduced-grazing, and grazing-prohibited treatments, respectively, whereas the differences between the G and G-KP micro-patches were 4.7–12.9%, 3.7–19.1%, and 2.6–14.4%, respectively. The similarity in SOM content between the G and G-KP micro-patches was greater than that between the K and G-KP micro-patches (Figure 3).

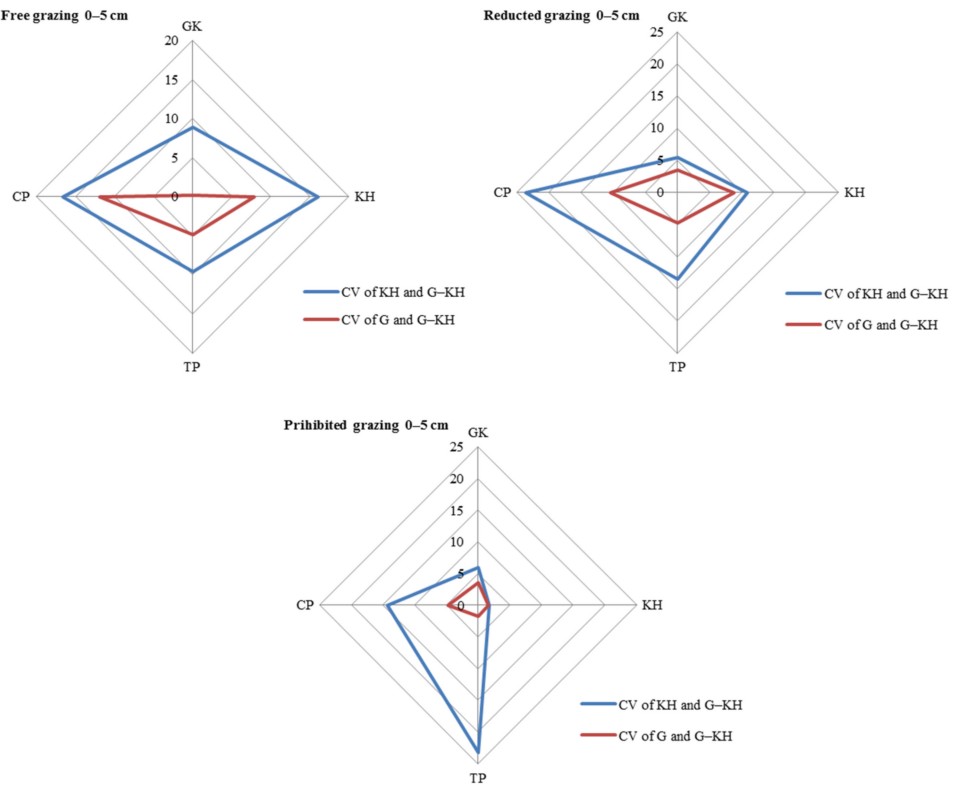

**Figure 3.** Coefficient of variation (CV) in soil organic matter content in the 0–5 cm layer between micro-patches in the degradation stages. Notes: CV of KH and G-KH: CV between the *Kobresia* and *Kobresia*-Gramineae transitional micro-patches; Free grazing 0–5 cm, Reduced grazing 0–5 cm, and Prohibited grazing 0–5 cm: CV in the free-grazing, reduced-grazing, and grazing-prohibited areas, respectively; GK, Gramineae-*K. humilis* meadow; KH, *K. humilis* meadow; TP, *K. pygmaea* meadow with mattic epipedon thickening; CP, *K. pygmaea* meadow with mattic epipedon cracking.

The CV between different micro-patches increased with increasing grazing intensity for most of the degradation stages. Taking the 0–5 cm layer as an example, the CV of KH and G-KH micro-patches increased from 6.0% to 8.9%, from 1.7% to 16.0%, and from 14.3% to 16.5% for GK, GH, and CP, respectively, with increasing grazing intensity. There was little difference in TP because the CVs of KH and G-KH micro-patches followed an opposite trend in this stage. This also applied to the CV of G and G-KP micro-patches as the grazing intensity increased (Figure 4).

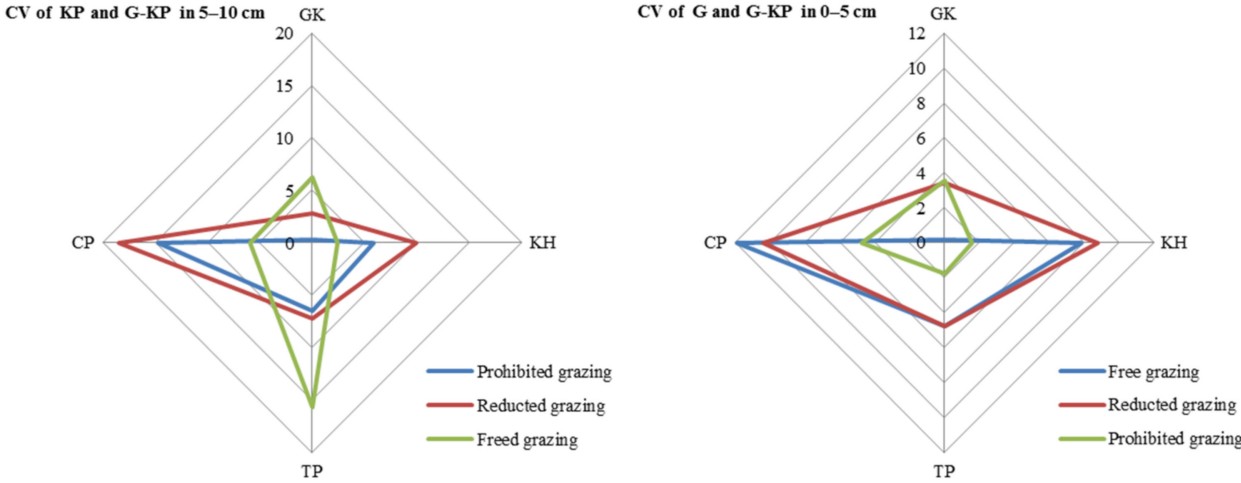

**Figure 4.** Variation in soil organic matter content between micro-patches in the plant community degradation process. Notes: CV of KH and G-KH: coefficient of variation at the 0–5 cm layer between the *Kobresia* and *Kobresia*-Gramineae transitional micro-patches, respectively; GK, Gramineae-*K. humilis* meadow; KH, K. humilis meadow; TP, *K. pygmaea* meadow with mattic epipedon thickening; CP, *K. pygmaea* meadow with mattic epipedon.

### 3.3. Variation in Soil Nutrient Content during the Plant Community Degradation Process

SOM content was one of the soil nutrient metrics that varied during the plant community succession process. The SOM content in the GK, CP, and TP stages decreased by 3.2–8.8%, 2.0–18.6%, 7.0–20.5%, and 10.2–24.2% in the 0–5 cm, 5–10 cm, 10–20 cm, and 20–30 cm soil layers, respectively, compared to the SOM content in the GH stage under the free-grazing, reduced-grazing, and prohibited-grazing conditions. The SOM content was 3.9–12.1% lower in the GK and CP stages than in the KH and TP stages in the 0–5 cm soil layer, but 3.5–18.5% higher than in the KH and TP stages in the 5–40 cm soil layer under the prohibited-grazing condition. The same phenomenon was observed under the reduced-grazing condition. These findings indicate that the SOM content in the GK and CP stages tended to increase in the mattic epipedon, but decreased in the deep soil layers under prohibited or reduced-grazing conditions (Figure 5).

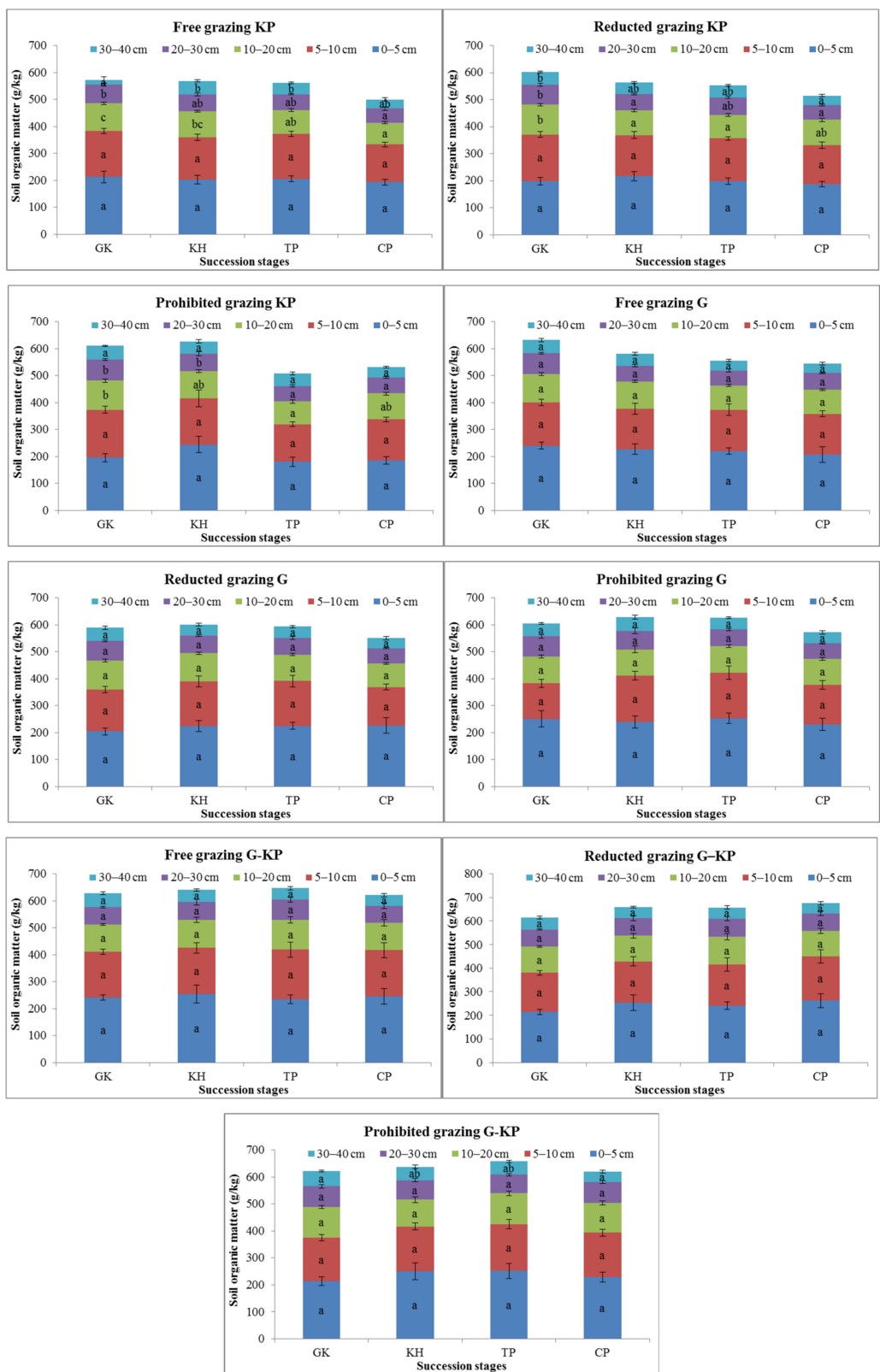

**Figure 5.** Variation in soil organic matter content in alpine *Kobresia* meadows. Notes: G, Gramineae micro-patches; G-KP, Gramineae transformation to *K. pygmaea* micro-patches; KP, *K. pygmaea* micro-patches; GH, Gramineae-*K. humilis* meadow; KH, *K. humilis* meadow; TP, *K. pygmaea* meadow with mattic epipedon thickening stages; CP, *K. pygmaea* meadow with mattic epipedon cracking stages. "a", "b", and "c" are the coefficients of significance for the different degradation stages.

### 3.4. Variation in avP in Soil during the Plant Community Degradation Process

avP is an available soil nutrient in the alpine *Kobresia* ecosystem. The avP content showed the same variation trend at the temporal, spatial, and degradation succession stages and the reduced-grazing intensity scale. Variation in the avP content in the G micro-patch mainly occurred in the mattic epipedon as the avP content in the 0–5 cm and 5–10 cm soil layers was only 5.3–27.4% and 2.9–27.1% higher, respectively, in the GK stage than in the other stages. However, the avP content was higher in the CP and TP stages than in the GK stage in the 0–10 cm soil layer (Figure 6).

The difference in avP content between the G and KP micro-patches showed the same trends in all plant community degradation succession stages. The avP content decreased by 28.1–62.0% and 16.2–37.4% in the 0–5 cm and 5–10 cm soil layers, respectively, in the GK stage compared with the TP stage. The difference in avP content between the G and G-KP micro-patches was less than the difference between the KP and G-KP micro-patches in the different succession stages (Figure 6).

G-KP micro-patches were mainly distributed in the cracking area. The avP content showed less variation in the G-KP micro-patch than in the other types of micro-patches in the degradation succession stages. The concentrations of avP in G-KP micro-patches followed the same variation with increased grazing intensity; the "back and forth" characteristics in one kind of micro-patch across different grazing intensities buffered the reduced soil nutrients in the ecosystem (Figure 6).

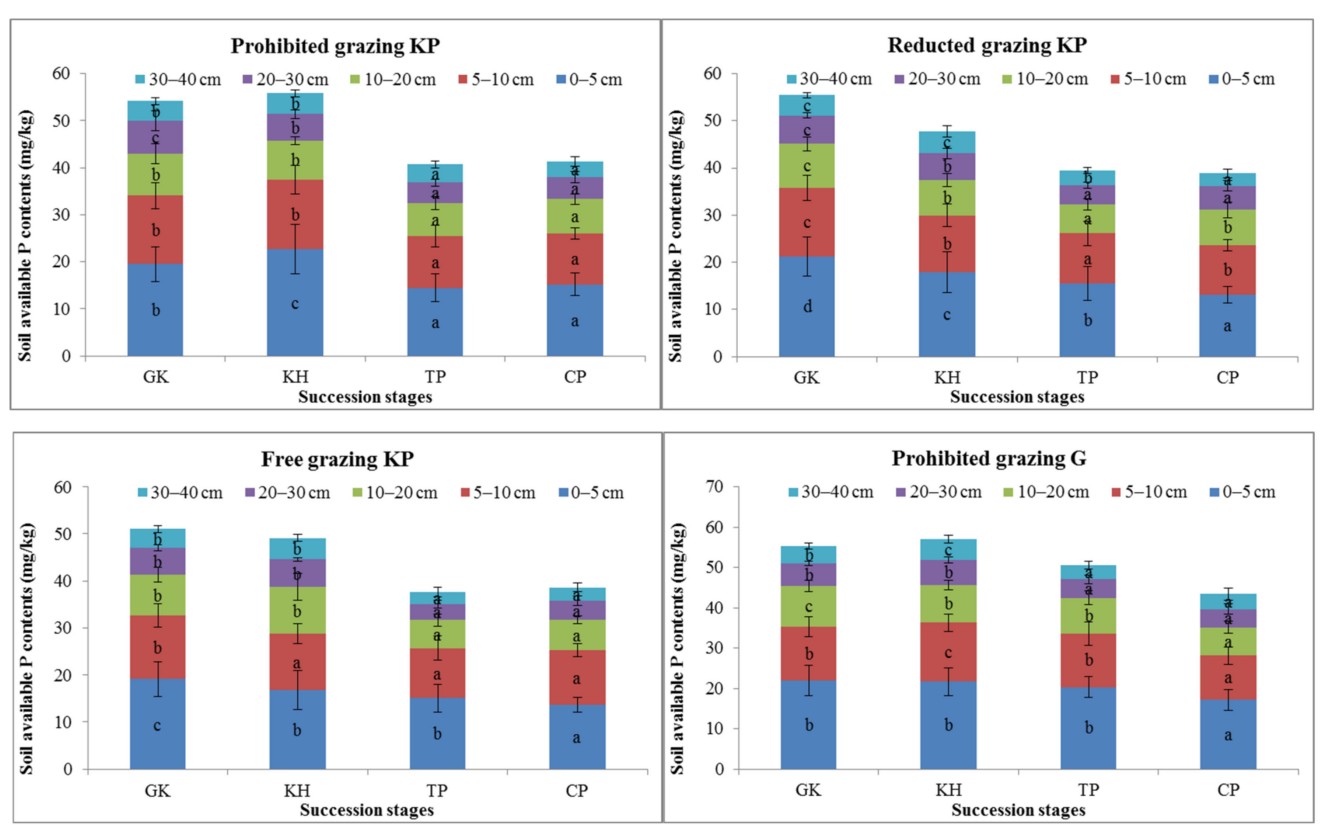

**Figure 6.** *Cont.*

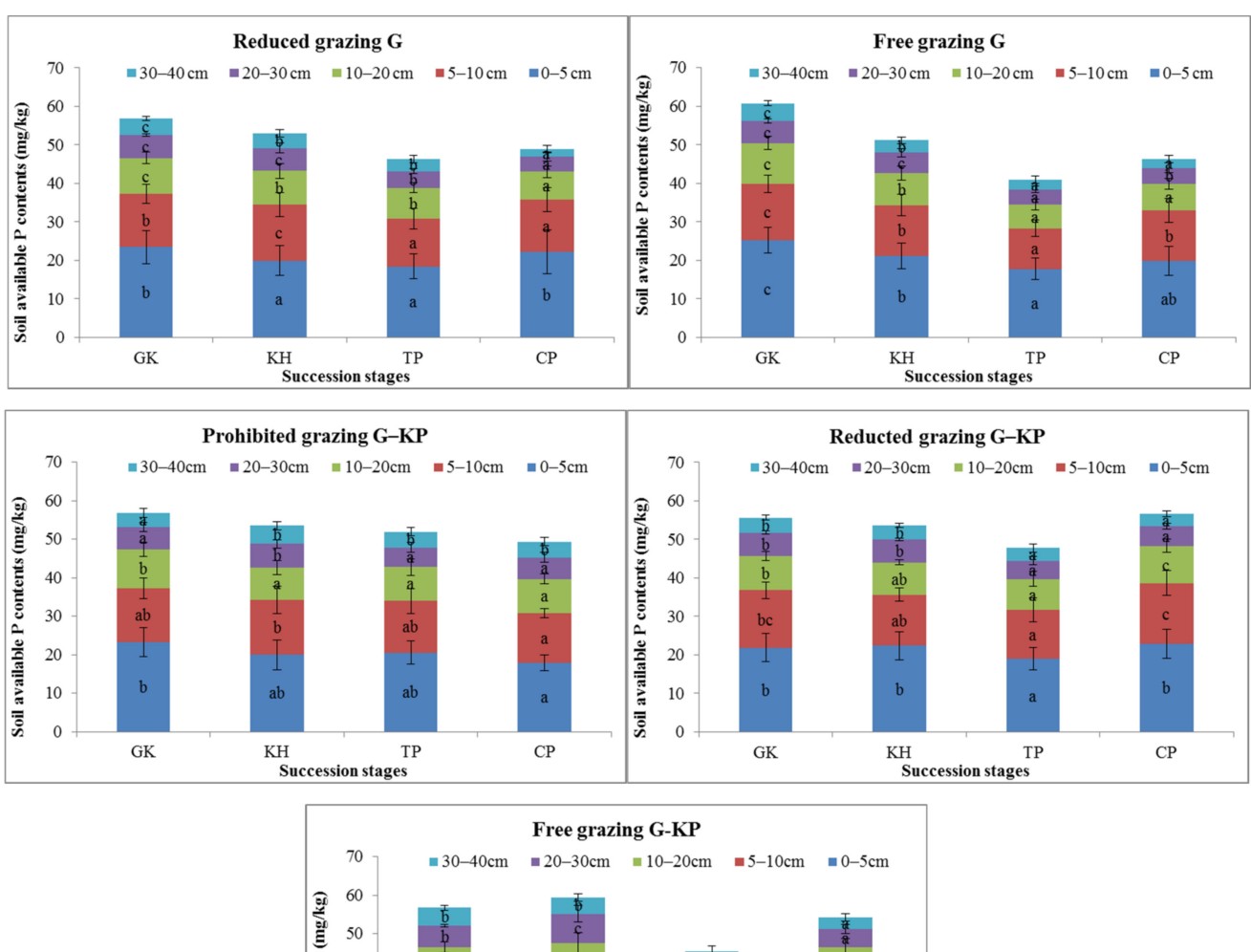

**Figure 6.** Variation in the available phosphorus content in alpine *Kobresia* meadows in different succession stages. Notes: G, Gramineae micro-patch; G-KP, Gramineae transformation to *K. pygmaea* micro-patch; KP, *Kobresia* micro-patch. "a", "b", and "c" are the coefficients of significance for the different degradation stages.

*3.5. Variation in avP in Soil across Micro-Patches*

The CV in avP content between G-KP and G micro-patches was lower than that between G-KP and KP micro-patches in most of the degradation stages and grazing intensity treatments (Figure 7).

The CV of avP content may decrease under medium-grazing-intensity conditions (reduced-grazing intensity treatment); we found that reduced-grazing intensity increased the similarity of the G and G-KP micro-patches, whereas decreasing the grazing intensity increased the similarity of the KP and G-KP micro-patches, except for TP and CP. This implied that decreasing grazing intensity could more easily increase the similarity of those micro-patches we studied in a relatively lowgrazing-intensity plot, whereas increasing variation in the micro-patches at relatively high grazing intensities increased their similarity to the plot examined before (Figure 8).

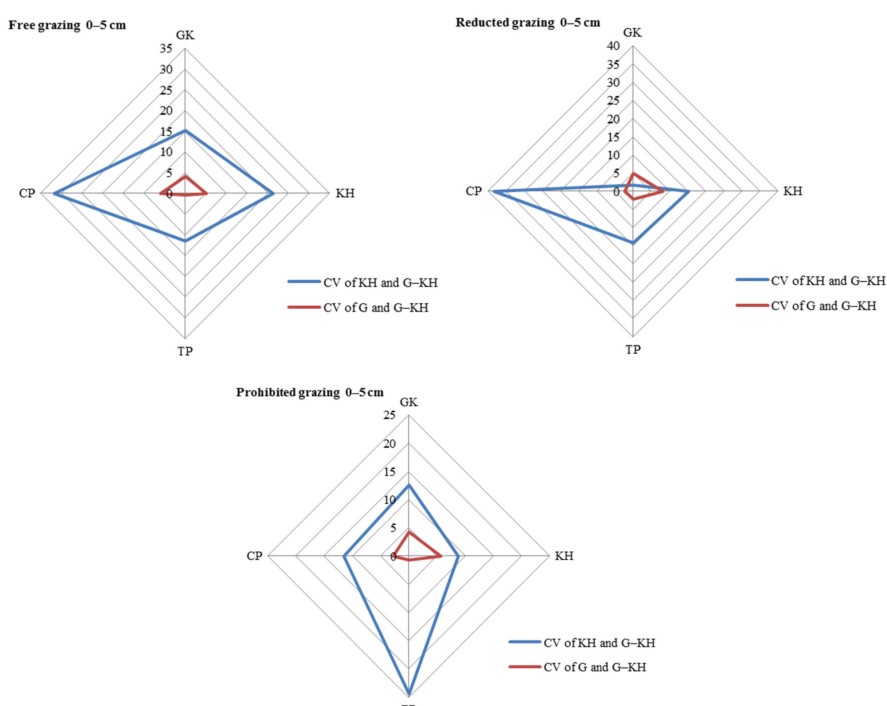

**Figure 7.** Coefficient of variation (CV) in soil organic matter content in the 0–5 cm layer between micro-patches in the degradation stages. Notes: CV of KH and G-KH: CV between the *Kobresia* and *Kobresia*-Gramineae transitional micro-patches; Free grazing 0–5 cm, Reduced grazing 0–5 cm, and Prohibited grazing 0–5 cm: CV in the free-grazing, reduced-grazing, and grazing-prohibited areas, respectively; GK, Gramineae-*K. humilis* meadow; KH, *K. humilis* meadow; TP, *K. pygmaea* meadow with mattic epipedon thickening; CP, *K. pygmaea* meadow with mattic epipedon cracking.

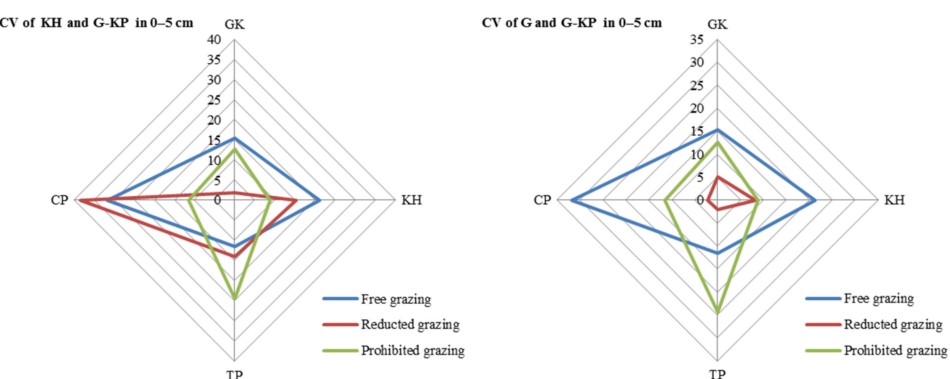

**Figure 8.** Variation in soil organic matter content between micro-patches in the plant community degradation process. Notes: CV of KH and G-KH: coefficient of variation at the 0–5 cm layer between the *Kobresia* and *Kobresia-Gramineae* transitional micro-patches, respectively; GK, Gramineae-*K. humilis* meadow; KH, *K. humilis* meadow; TP, *K. pygmaea* meadow with mattic epipedon thickening; CP, *K. pygmaea* meadow with mattic epipedon.

### 3.6. Variation in avP in Soil at the Temporal Scale

The variation in soil avP content was lower under the free-grazing conditions than uner the prohibited- or reduced-grazing conditions.

The avP content of the soil showed significant temporal variation. The temporal CV of avP content in the GK, KH, TP, and CP stages was 31.5%, 29.4%, 39.4%, and 32.7%, respectively, under the free-grazing conditions and 32.1%, 34.1%, 36.8%, and 32.2%, respectively, under the prohibited-grazing conditions. These results indicate that, when the grazing

intensity increased to a certain extent, the spatial heterogeneity of the soil avP content also increased (Figure 9).

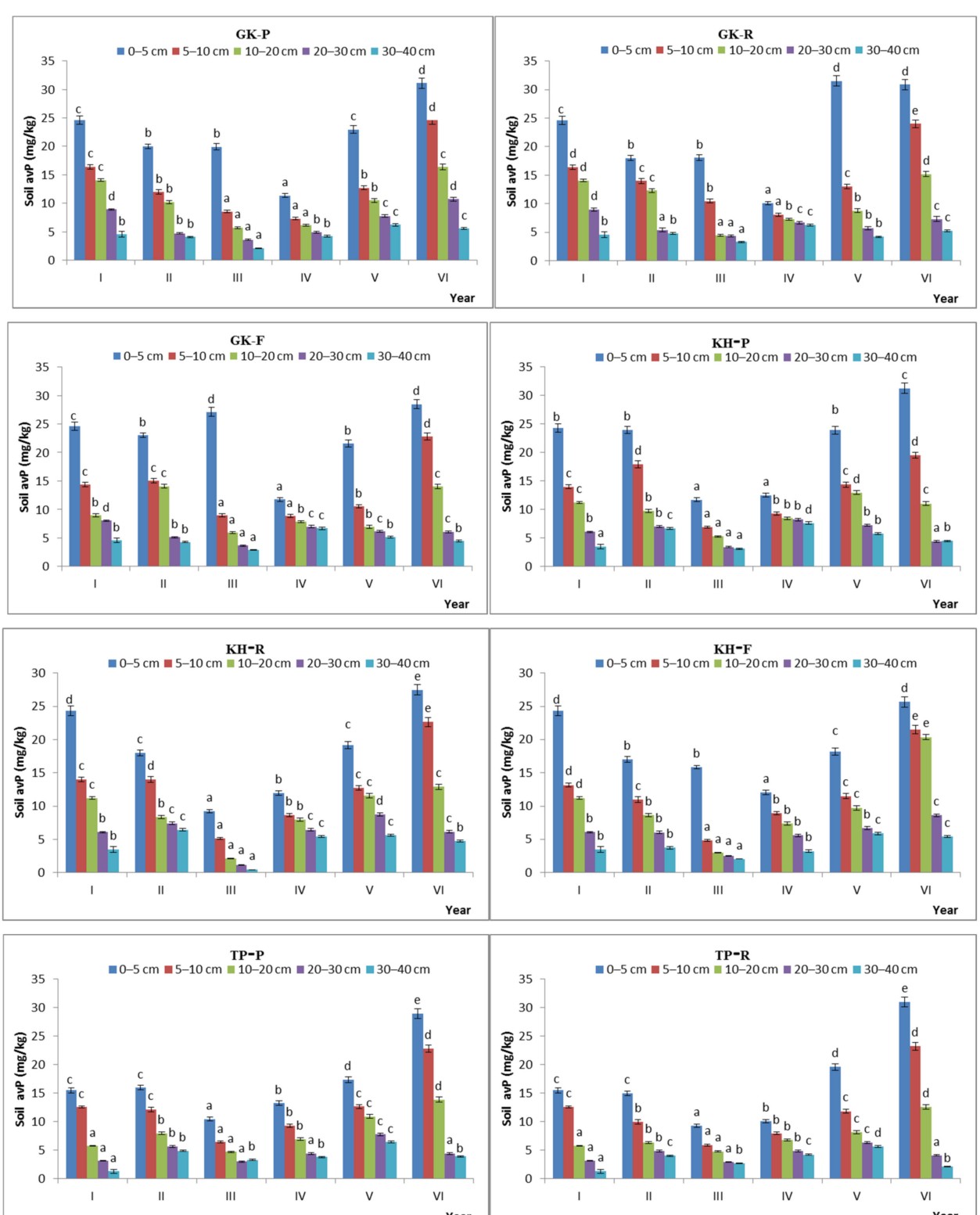

**Figure 9.** *Cont.*

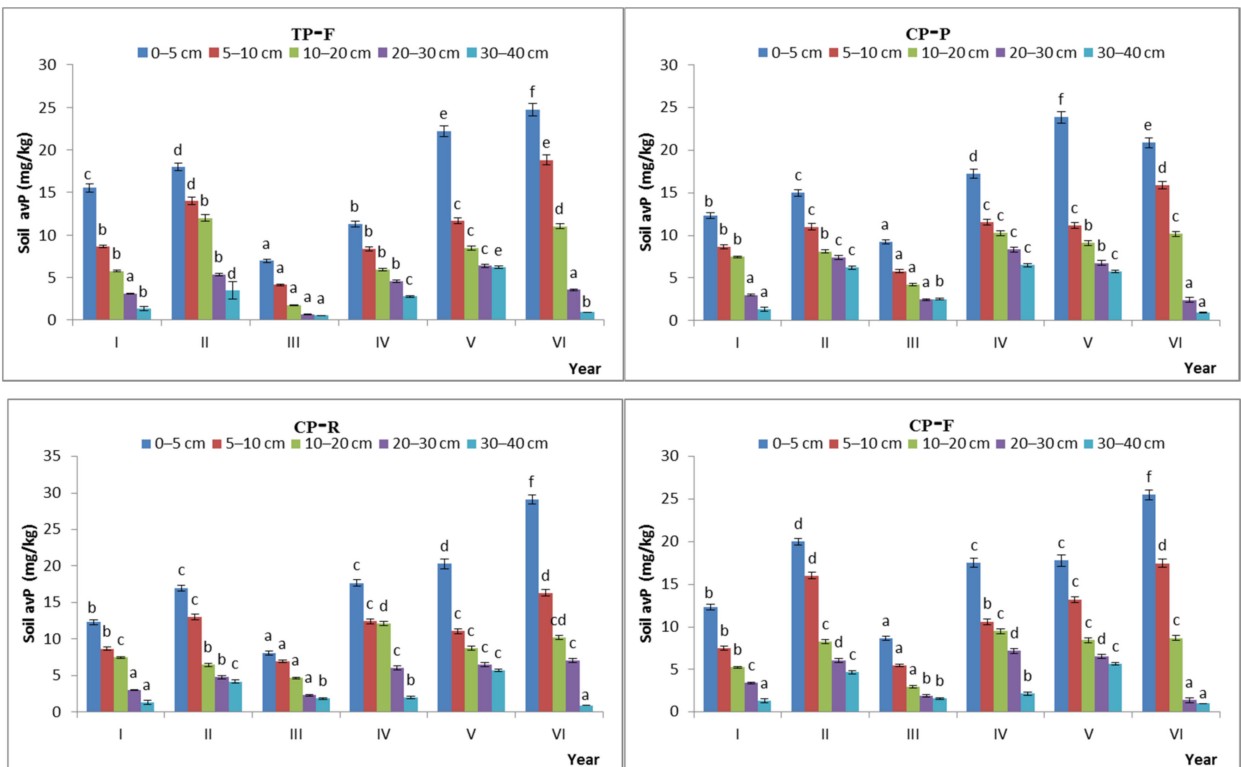

**Figure 9.** Characteristics of available phosphorus content during the natural recovery process in different steady-state alpine *Kobresia* meadows. Note: P: grazing-prohibited plot; R: reduced-grazing plot; F: free-grazing plot; GK, KH, TP, and CP: Gramineae-*K. humilis* meadow, *K. humilis* meadow, *K. pygmaea* meadow with mattic epipedon thickening, and *K. pygmaea* meadow with mattic epipedon cracking, respectively; I, II, III, VI, V, and VI, indicate 2013, 2014, 2015, 2016, 2017, and 2018, respectively; "a", "b", "c" and "d" are the coefficients of significance for different years.

## 4. Discussion

### 4.1. Overgrazing Could Trigger Synchronized Changes in the Plant Community and Soil Nutrients in Micro-Patches, Which May Increase the Risk of Regime Shift across Different Plant Communities in the Ecosystem Succession Process

Based on a multivariable analysis of the characteristics of soil nutrients in micro-patches, we found that the spatial scale was the dominant scale affecting the variation in total nutrient content of the soil, whereas the temporal scale was the main scale affecting the variation in available nutrient content of the soil. A comparison of variation in the degradation and recovery processes showed that most of the soil nutrients were more affected by the degradation process resulting from overgrazing, while others were more affected by the recovery treatment scale. Preliminary studies have shown that the accumulation and consumption patterns of soil nutrients are closely related to the characteristics of the plant communities [1,3] and that the strength and rate of soil nutrient reactions differ depending on variations in the plant communities [33]; this may explain the asynchrony in different types of soil nutrients in overgrazing and recovery conditions. In addition, we found one characteristic of soil nutrients in one type of micro-patch that differed between the degradation process stages and recovery conditions in the alpine *Kobresia* meadow. For example, the SOM content was more similar between the G-KP and G micro-patches than between the G-KP and KP micro-patches, even though the G-KP micro-patches originated from the KP micro-patch. However, the soil nutrient content of the G micro-patches was significantly higher than that of the KP micro-patch, and increasing grazing intensity increased the variation between the two types of micro-patches. This phenomenon was observed in the plant community degradation process induced overgrazing and recovery processes. The SOM content was significantly higher in the G micro-patches than in the KP

micro-patches in conditions of reduced-grazing intensity, indicating that the degradation in plant community could affect the nutrient accumulation and storage patterns of soil from the same micro-patches. Consequently, the soil nutrient content of the entire ecosystem changed as the plant community changed because the G micro-patches lost their dominant position in the overgrazing ecosystem and the total soil nutrient content inevitably decreased [4]. Moreover, the regulation of soil nutrient content of the ecosystem was evident from the amount of available soil nutrients, such as the avP content. Therefore, overgrazing in ecosystems could have a positive effect on the characteristics of plant communities and their nutrient contents; if positive feedback became the mainstream feedback, then the degradation trend would be unstoppable due to the emergence of G-KP micro-patches adapted to a high soil nutrient content, and this regulation might be considered as the evidence for the self-stabilization of alpine *Kobresia* meadows under the overgrazing.

### 4.2. Different Types of Micro-Patches in Alpine Kobresia Meadows Experience Different Feedback Effects during the Degradation and Recovery Processes, Which Help the Ecosystem Maintain Relative Stability

From the perspective of the plant community degradation process, the GK stage is the grazing climax community in the alpine *Kobresia* meadow degradation process [4,34], the *K. pygmaea* meadow stage is the disclimax community [4], and TP and CP are the typical sub-succession stages in *K. pygmaea* meadows under overgrazing conditions [3]. The SOM content showed no significant differences between succession stages in the topsoil layer, but was significantly higher in the GK stage than in other stages in the deep soil layers. Therefore, the storage of SOM in the soil profile and amount of available nutrients in the soil changed under conditions of overgrazing. Previous studies have shown that the degree of spatial homogeneity between different succession and sub-succession stages changes markedly with increasing grazing intensity [4]. This may occur because of variations in different soil nutrients between different micro-patches and in the nutrient ratios of the micro-patches. The SOM content decreased with increasing grazing intensity prior to the TP stage (*Kobresia* meadow with mattic epipedon thickness), and then in the CP stage (*Kobresia* meadow with mattic epipedon cracking). As the ratio of G-KP micro-patches to other micro-patches increased, the soil nutrient content increased, which in turn increased the heterogeneity of the ecosystem, and led a "back and forth" trend along with increasing grazing intensity, which may be considered self-stabilization with multiple stable states [35]. Thus, the decrease in the ratio of G micro-patches to other micro-patches was a positive feedback mechanism that decreased the SOM content in response to increasing grazing intensity, while an increase in the number of G-KP micro-patches compensated for the decrease in the number of G micro-patches. This regulatory mechanism helped prevent the continuous decrease in soil nutrient content, provided additional soil nutrients to support vegetation restoration, and provided negative feedback for the reduction in the primary productivity of the pasture due to overgrazing.

This negative feedback effect also occurred in the recovery treatment process. For example, the recovery treatment (prohibited-grazing treatment) did not increase the SOM content in the GK stage, but did in the TP stage. Previous studies have shown that the GK stage has the highest SOM content, but reducing or prohibiting this type of pasture may not increase SOM storage; thus, it may be considered negative feedback to the recovery process. However, recovery treatment carried out in the CP or TP stages may increase the SOM content and be considered positive feedback to the recovery process. The different stages showed very different responses to the same types of environmental change, with the final results showing that the soil nutrient content was maintained within a relatively fixed threshold range. However, relatively stable soil nutrient content and structural characteristics also form the basis of relatively stable vegetation communities.

The question of why the soil nutrient content of the alpine meadows was maintained within a relatively small range across different stages remains unanswered. This effect may have arisen due to a co-evolutionary relationship between the plants, soil, and livestock during long-term livestock husbandry in the QTP [36]. Traditional nomadism uses the

management strategy of "short-term high-intensity grazing and long-term recovery for animal husbandry" [37]. Short-term, high-intensity grazing in an alpine meadow may result in long-term changes to the components and structure of the ecosystem [38]. In such conditions, the alpine meadow must adopt a series of positive and negative feedback mechanisms to resist external interference and maintain the balance of the system. TP and CP are the disclimax zonal vegetation communities of Gramineae-*Kobresia* meadows in overgrazing conditions [3]. During the degradation process, TP and CP displayed the ability to buffer the decrease in nutrients caused by overgrazing. Thus, they may be considered nutrient-regulating buffer zones in the degradation succession of Gramineae-*Kobresia* meadows.

### 4.3. K. pygmaea Meadows Are the Final Defense against the Effects of Overgrazing on Primary Production in the Alpine Meadow Ecosystem

The alpine *K. pgymaea* meadow is the most widespread *Kobresia* meadow in alpine pasture ecosystems [39]. One of the reasons for the *K. pgymaea* meadow becoming the dominant disclimax zonal vegetation community in the Gramineae-*Kobresia* meadow degradation process [4] is the two-way feedback system used for the adjustment of soil nutrients in micro-patches. Alpine meadow soil depths >5 cm are dominated by *K. pgymaea* micro-patches. This type of alpine meadow is subject to a large range of grazing intensities [3]. The characteristics of these meadows are low soil nutrient inventory and forage production, which lead to slow growth and energy utility in the ecosystem as adaptations to high-intensity livestock grazing [1].

Mattic epipedon cracking as a result of heavy grazing in *Kobresia* meadows is a regulatory mechanism that allows ecosystem recovery. These types of micro-patches can buffer against the degradation process caused by overgrazing, which is characterized by decreases in plant community production or soil nutrient content. This phenomenon may postpone succession from a *K. pgymaea* meadow to black-beach soil and maintain the primary production the *K. pgymaea* meadow in relative stable conditions over a long period of time [1]. Thus, we infer that the *K. pgymaea* meadow, in conditions of livestock overgrazing with the greatest number of *K. pgymaea* micro-patches, forms the last line of defense against the *K. pgymaea* meadow degradation succession process. The buffer mechanism of mattic epipedon cracking may increase the available nutrient content of the soil, and even the total soil nutrient content, thereby improving the existing plant community and increasing the amount of biomass. Thus, mattic epipedon cracking is considered a strategy used by the plant community for production recovery in an ecosystem [1,3].

However, mattic epipedon cracking in alpine meadows increases the risk of soil and water loss from the surface of the ecosystem [4]. If G-KP micro-patches are exposed to heavy grazing, then the recovering plant community may be prioritized for livestock grazing as these plants are edible and present in greater quantities in G-KP micro-patches than in KP micro-patches. However, this would increase the risk of soil bareness and cracking in the mattic epipedon, which may increase the risk of soil being carried off by water, the weathering of soil, and widening of cracks by water erosion. The final results of these serious effects are erosion of the mattic epipedon, replacement of the topsoil layer with soil parent material, restarting of the zonal vegetation succession process, establishment of annual and biennial plant communities, degradation of the alpine meadow into black-beach soil, and disappearance of livestock husbandry [3]. Thus, the *K. pygmaea* meadow is in the regime shift stage in the recovery and degradation processes, which can be considered a key management stage in maintaining the stability of the ecosystem in overgrazing conditions and resisting further degradation.

## 5. Conclusions

Micro-patches in alpine *Kobresia* meadows were found to vary in terms of soil nutrient characteristics. The main results of this study are as follows: (1) the nutrient content of micro-patch soil decreased in the order G-KP transition micro-patch >G micro-patch

>KP micro-patch; (2) the variation across different micro-patches showed that the level of heterogeneity first decreased and then increased; and (3) the recovery process increased the heterogeneity of soil nutrients during heavy grazing stages, but decreased heterogeneity in the presence of low grazing intensity. Thus, we concluded that overgrazing decreases the soil nutrient content, decreases the fertilizing island effect of the Gramineae plant community, and decreases the total nutrient content of the soil to support plant community growth. However, cracking in the mattic epipedon increases the soil nutrient content, but puts the alpine meadow at risk of erosion and degradation into black-beach soil. Thus, cracking in alpine meadows is the last line of defense against the alpine *Kobresia* meadow degradation process and may be considered the beginning of recovery after the degradation process.

**Author Contributions:** Conceptualization, G.C. and X.G.; methodology, B.F.; software, B.F. and J.H; validation, L.L.; formal analysis, L.L.; investigation, D.Q., Q.L., B.F., B.L., Y.L., M.S., Y.D. and J.H.; resources, G.C.; data curation, L.L.; writing—original draft preparation, L.L.; writing—review and editing, L.L.; visualization, L.L.; supervision, G.C.; project administration, G.C.; funding acquisition, L.L. All authors have read and agreed to the published version of the manuscript.

**Funding:** This research was funded by the Natural Science Foundation of Qinghai Province (2020-ZJ-720).

**Institutional Review Board Statement:** Not applicable.

**Data Availability Statement:** The data presented in this study are available in the article.

**Acknowledgments:** We are grateful to Dong Shikui from Beijing Forestry University for proposing the article and writing the paper.

**Conflicts of Interest:** The authors declare no conflict of interest.

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
