# Peer review of "The Process of Soil Nutrient Stabilization in Micro-Patches in Alpine Kobresia Meadows"

_diversity, doi:10.3390/d14080656_

Round 1
Reviewer 1 Report
Attached

Author Response
dear professor
I am so sorry to return the manuscript so late. I give the point by point responds to the comments, and I sorry, those days we are in the work of field research, so if I did not good as you require, please contact me at any time
best wishes
Li

Reviewer 2 Report
The manuscript reports an important research on natural systems. The manuscript is well written and contains adequate amount of data, well analysed and nicely presented. Some minor mistakes were found as shown below: however, list can be long. Pl read the MS carefully make it mistake free.
1. Lines 22-23: mention % only once at last in the series of per cent values;
69.8–79.7, 61.4–80.35, 49.8–79.58, 60.52–22 76.34, 46.44–89.89, 45.5–71.36, and 59.21–65.38%,
2. Also give % values upto 1 digit after decimal through out
3. Line 29 to 35: very long and confusing sentence, break this into smaller sentences
4 Line 32: Delete 'it' mentioned before between...
5. Line 86: give space after ..state
6. Line 106: Give space after process, check whole Manuscript for such inadvertent cases.
7. Mention symbols for nutrients like carbon (C), nitrogen (N), phosphorus (P) and for other nutrients through out after mentioning full name once in the begining. This will reduce length of paper
Line 346: Delete 'a' before medium, there are some more such mistakes, pl read the MS thoroughly
Correct line 347 also... reduction grazing intensity --
Easily?? use proper word .. may be "'easy'
Overall, the research reported has enough merit.
Author Response
Dear professor
Thank you for your suggestion, I am so sorry to return too late, because those days we are carrying out the field research. But if there are any question in the manuscript, pleaes contact me at any time, I will deal with them at the first time . by the way , I give the revised manuscript as the attachment.
Beat wishes
yours Li

Reviewer 3 Report
The authors present an interesting work carried out in alpine ecosystems where the research work is limited. One aspect is that the experiment is very complex, including, three type of patches, time repeated observations (six years), four degradation levels and three recovery treatments. Setting treatment and plots in the field is always difficult, especially in ecological research because we do not have true replicates as we could have in a laboratory or nursery experiment.
This mentioned complexity of the study makes sometimes difficult to follow the results of the work. Indeed, I think that the discussion section needs deeper explanations for the more relevant results. This manuscript is publishable but before that, need some work.
It is confusing the term available K. The authors need to clarify this if they are talking about K as exchangeable K. What the author were expecting from total determinations of total K (TK) and how this may be connected to the studied processes. There is a similar question for total P and available P. The authors do not cite proper references so that the reader can find the lab methods used.
I am attaching a PDF file with many questions and suggestions for the authors, and I hope all of them help to build a new version of the manuscript.

Author Response
Dear professor
I am so sorry to reture the manuscript so late. Because those days we are do the field research, if the manuscript have any question, please contact me at any time
Thank you
yours Li

Round 2
Reviewer 3 Report
The mnaucript can continue with the editoril process. I found this points to attend.
I strongly recommend a revision of the English for this manuscript.
SECOND REVIEW
Line 33, replace “the” by “a”.
Lines: 152-154, can be changed to:
In general, we can recognize the micro-patches by plant community more easily than their soil nutrient status, so most of the time we pay a less attention on soil nutrients bel- lowed the plant community.
Line 171-174, , can be changed to:
We hypothesize that the soil nutrient contents and their stabilization in typical mi-cro-patches vary both in spatial and temporal, according to the degradation and recovery process scales, and the factor that has the greatest variation to the total variation in soil nutrients is the key factor causing variation
Line 257: Should say this?
(Variation coefficient in %)
Line 276
…. were more sensitive ……..
Line 292
Replace “In the research” by “This research showed that ….”
Line 303
Replace “ In” by “At”
Line 304
Replace “ as the example” by “for example”
Line 321
Delete “of”
Line
Replace “waved” by “followed the same…”
Author Response
Dear professor
Thank you for your suggestions. I had modified the manuscript, and they had been high light labeled in the paper. I know I am poor in English writing, and I want to revise the article by the native speaker when the article have no problem in logical and other mistakes, so could you give me the chance to revise the article by that time. Thank you.
best wishes
yours Li
